# Extraction of Graphene’s RF Impedance through Thru-Reflect-Line Calibration

**DOI:** 10.3390/mi14010215

**Published:** 2023-01-14

**Authors:** Ivo Colmiais, Vitor Silva, Jérôme Borme, Pedro Alpuim, Paulo M. Mendes

**Affiliations:** 1CMEMS—Center for Microelectromechanical Systems, University of Minho, Campus de Azurém, 4800-058 Guimarães, Portugal; 2INL—International Iberian Nanotechnology Laboratory, Av. Mestre José Veiga, 4715-330 Braga, Portugal; 3Center of Physics, University of Minho, Campus de Gualtar, 4710-057 Braga, Portugal

**Keywords:** quantum capacitance, kinetic inductance, graphene, 2D materials, TRL calibration, RF

## Abstract

Graphene has unique properties that can be exploited for radiofrequency applications. Its characterization is key for the development of new graphene devices, circuits, and systems. Due to the two-dimensional nature of graphene, there are challenges in the methodology to extract relevant characteristics that are necessary for device design. In this work, the Thru-Reflect-Line (TRL) calibration was evaluated as a solution to extract graphene’s electrical characteristics from 1 GHz to 65 GHz, where the calibration structures’ requirements were analyzed. It was demonstrated that thick metallic contacts, a low-loss substrate, and a short and thin contact are necessary to characterize graphene. Furthermore, since graphene’s properties are dependent on the polarization voltage applied, a backgate has to be included so that graphene can be characterized for different chemical potentials. Such characterization is mandatory for the design of graphene RF electronics and can be used to extract characteristics such as graphene’s resistance, quantum capacitance, and kinetic inductance. Finally, the proposed structure was characterized, and graphene’s resistance and quantum capacitance were extracted.

## 1. Introduction

Two-dimensional (2D) materials have attracted researchers due to their unique properties. Their optical, electronic, and mechanical properties indicate that these materials have great potential for novel applications in electronics and optoelectronics [1,2,3,4]. An example of such material is graphene, which is a 2D material that is constituted by carbon, structured in a one-atom-thick honeycomb layer. Graphene is an attractive material for electronic and radio frequency (RF) applications due to its high carrier mobility and high current density [2,5,6,7]. These characteristics make it an attractive material for RF passive components such as inductors and capacitors. Through graphene’s kinetic inductance and high conductivity, it claims an improvement of Q-factors and an increase of inductance density values, thus reducing the total area of integrated circuits [2,5,8,9,10,11]. Furthermore, graphene capacitors can make use of their tunability to enable variable capacitors that are controlled through direct current (DC) biasing, making use of its quantum capacitance [2,12,13,14]. However, due to the particular characteristics and novelty of 2D materials, such as graphene, the methods for extraction of the required design properties at RF frequencies are still under development and assessment.

To perform RF characterization, the first mandatory step consists of the vector network analyzer (VNA) calibration to shift the measurement plane to the device under test (DUT). That can be achieved through several available alternatives such as short-open-load-through (SOLT), line-reflect-match (LRM), and TRL [15]. The TRL calibration algorithm, when coupled with precision transmission lines, provides one of the highest calibration accuracies [16,17]. It relies on the characteristic impedance of the designed transmission lines, on two transmission lines with different lengths, and one-port reflect termination [18]. Furthermore, for the measurement of graphene devices, it was demonstrated in [15] that not only is TRL the best candidate for such measurements, but the authors also claim that using SOLT could lead to erroneous values of graphene properties. Since the commercial SOLT solutions do not remove the contributions of the transmission lines even though the resistance of the 2D material can be extracted with small error (since the 2D material’s resistance is much larger than the transmission line’s resistance), the small inductance and capacitance values may have a similar order of magnitude, from which these errors may arise.

In this work, the issues of implementing the TRL calibration applied to the characterization of 2D materials, such as graphene, is presented. The impact on the RF parameter extraction due to the design of the calibration structures, such as contact thickness, substrate loss, contacting graphene, and backgate design, are analyzed. Finally, resistance and quantum capacitance are extracted for graphene and processed using standard commercial 8-inch wafers processing equipment.

## 2. TRL Calibration Implementation

This section discusses the implementation of the TRL calibration. It focuses on the design of the calibration structures, their frequency span, and the characteristics of the substrate. In this case, coplanar waveguides were used so that it is possible to contact them with ground signal ground RF probes and fabricate the device in small dimensions so that it can be used with graphene. Then, the determination of the characteristic impedance is discussed since the calibration is related to the impedance of the transmission lines. Finally, the extraction of the parameters used in this work is presented.

### 2.1. Design of Calibration Structures

To explore this calibration, it was necessary to define the calibration structures as well as the device under test. The number of lines used depends on the frequency span for which the calibration is used. The optimal line length is ¼ wavelength, and each thru/line pair has a usable bandwidth of 8:1 (frequency span/start frequency) [16,19]. Since 3-line standards cover a bandwidth of 512:1, for this calibration spanning from 1 to 65 GHz, 3-line standards are required. To determine the length of the lines, a geometric segmentation of the bands can be used. For the three-line standards, the crossovers FT1 and FT2 are:(1)$$f_{T1}=f_{L}*10^{\log\left(\frac{f_{H}}{f_{L}}\right)*\left(1/3\right)}$$
(2)$$f_{T2}=f_{L}*10^{\log\left(\frac{f_{H}}{f_{L}}\right)*\left(2/3\right)}$$
where $f_{Tn}$ is the crossover frequency, $f_{L}$ is the lower frequency, and $f_{H}$ is the higher frequency. Then, the center frequency is determined through the arithmetic average of the crossover points. To determine the length of each line, since the optimum is one-fourth of the wavelength, the dielectric constant is used:(3)$$l=\frac{c}{f_{C}\sqrt{\varepsilon_{r}}}*\frac{1}{4}$$
where c is the speed of light, $f_{C}$ is the center frequency of the line standard, and $\varepsilon_{r}$ is the effective relative permittivity.

To develop the calibration structures, we considered that the measurement probe would have a ground signal ground configuration with a pitch of 100 µm, and the calibration frequency would be from 1 GHz to 65 GHz to validate the calibration from low GHz up to mm-wave. The substrate was silicon, 700 µm thick, with a 100 nm silicon oxide layer, a typical oxide thickness that allowed us to fabricate a buried gate. The silicon had a dielectric constant of 11.9 and a resistivity of 10 × 10^3^ Ω.cm, and the silicon oxide had a dielectric constant of 3.9 and loss tangent of 0.01. The metal used was copper, with a conductivity of 5.8 × 10^7^ S/m. The dimensions (Figure 1) of the coplanar waveguide (CPW) were calculated so that a 50 Ω impedance line was obtained: 50 µm signal width, spacing of 30 µm, and ground width of 90 µm. The length of the THRU was 350 µm to allow for an easier measurement probe placement and prevent crosstalk, equaling a 175 µm line for the measurement of the remaining structures. The calibration frequency was from 1 GHz to 65 GHz; however, to avoid resonance at higher frequencies in the longer lines and determine the line lengths, we considered that the frequency range was from 1 GHz to 70 GHz to shorten the lines and shift possible resonant frequencies outside the calibration range. Since the predicted effective permittivity of the substrate is 6.36, the length of the lines was 11,614 µm, 2818 µm, and 683 µm. These calibration structures are presented in Figure 1 and are labeled THRU, REFLECT, and LINE. It was considered that the device under test (DUT) occupied an area of 100 µm × 50 µm, allowing for a large area to build a device. In this case, it also allows us to characterize and extract the sheet parameters of graphene.

These calibration structures are tested to assess their viability for device parameter extraction, as well as their application on the measurement of graphene. Furthermore, some parameters, such as substrate conductivity, metal thickness, and inclusion of graphene-specific features, are studied.

### 2.2. CPW Characteristic Impedance Measurement

The result from the TRL calibration involves the characteristic impedance of the transmission lines. Therefore, the S-parameters reference characteristic impedance and in most cases, should be converted to the typical 50 Ω reference impedance. To determine the characteristic impedance of the tansmission line, the procedure proposed in [20] was undertaken. A previous calibration of the measurement setup, for example SOLT, must be performed to shift the reference plane to the beginning of the transmission lines. Afterwards, the TRL calibration can be performed, which allows us to extract the fabricated transmission lines’ characteristic impedance. When performing this calibration, the TRL calibration error terms represent the S-parameters of the transmission lines.

Three values obtained from the scikit-RF package [21] are used to build the S-parameter matrix and extract the characteristic impedance: forward reflection tracking (*FRT*), forward directivity (*FD*), and forward source match (*FSM*). It is now possible to obtain the S-parameter matrix [20]:(4)$$S_{CPW}=\left[\begin{array}{cc} FD & \sqrt{FRT}\\ \sqrt{FRT} & FSM \end{array}\right]$$

It is now possible to extract the characteristic impedance of the lines by transforming the S-parameters to T-parameters and through Equations (5) and (6) [20]:(5)$$G=\sqrt{\frac{T12+T21^{2}}{4+T12+T21^{2}}}$$
(6)$$Z=50\frac{1+G}{1-G}$$

This method, however, is only accurate at lower frequencies. To obtain a more accurate approximation, the propagation constant that is given by the TRL calibration can be used by solving the capacitance and substrate conductance per unit length:(7)$$C=\mathrm{real}\;\left(-\frac{j\gamma }{2\pi fZ}\right)$$
(8)$$G=\mathrm{real}\;\left(\frac{\gamma }{Z}\right)$$

The capacitance and conductance values are obtained from the lower frequency results of Equations (7) and (8), respectively, given that the method is more accurate in this range [20]. Since the propagation constant is given by the TRL calibration, it is now possible to determine the characteristic impedance through Equation (6). Since this impedance has been determined, it is now possible to calibrate and renormalize the TRL calibration results to 50 Ω by using the characteristic impedance values in the calibration procedure.

### 2.3. DUT Parameter Extraction Methodology

The goal is to extract the series resistance, quantum capacitance, and kinetic inductance of graphene, since that information is necessary to develop interconnects, components, devices, and systems. Depending on the device geometry and number of graphene layers used, it is expected that the graphene device can have a capacitive or inductive behavior. To perform the TRL calibration, the NISTMultilineTRL procedure from the scikit-RF package [21] was used. After performing this calibration to extract the parameters, i.e., calculating the inductance (L) or capacitance (C) and resistance (R) of the DUT, the Y-parameters were used. To obtain the desired values, the real and imaginary parts of the $Y_{2,1}$ of the DUT were used:(9)$$L=-1\times 10^{9}\times \frac{\mathrm{imag}\left(\frac{1}{Y_{2,1}}\right)}{2\pi f}$$
(10)$$C=\frac{1\times 10^{12}}{2\pi f*\mathrm{imag}\left(\frac{1}{Y_{2,1}}\right)}$$
(11)$$R=-\mathrm{real}\left(\frac{1}{Y_{2,1}}\right)$$

## 3. Two-Dimensional Materials TRL Calibration Methodology Performance Assessment

The proposed calibration method is carried out and tested in this section. Then, it is used to demonstrate different factors that can negatively impact or even prevent accurate measurements from being obtained. These are the metal thickness of the calibration structures and the substrate loss. Then, a calibration with thick copper contacts and high resistivity substrate is demonstrated, as well as including thin gold contacts and a buried gate for the characterization of graphene. Since the inclusion of the gold contacts and the buried gate are a modification of the original TRL calibration, their influence on the characteristic extraction is reported.

The procedure starts with a typical calibration (such as SOLT) of the measurement system to shift the measurement planes to the transmission lines of the TRL calibration. Then, the TRL calibration structures are measured, and their characteristic impedance is extracted through the steps described in the previous section. In this case, the simulation software is used to simulate only the calibration structures, mimicking the initial step (SOLT and structure measurement for characteristic impedance extraction). After obtaining the characteristic impedance of the transmission line, the calibration can be carried out without any pre-calibration of the VNA, and the desired RF characteristics can be extracted. This corresponds to including two 2 m long, 53 Ω coaxial cables in the simulation, one for each measurement port. This demonstrates that the calibration can be performed even with slightly mismatched coaxial cables.

Implementing this calibration with 2D materials can be a challenging task due to their physical dimensions and characteristics. Another constraint related to characterizing two-dimensional materials is their inherent high series resistance. Graphene has a sheet resistance of 0.5 kΩ to 1.5 kΩ [22]. Since the resistance of the device being measured is comparable or higher than the impedance between the measurement contacts (capacitance between both contacts), it is necessary to remove it. A similar structure, without the DUT, can be used to determine this capacitance, subtract it from the device’s measurements, and correct the obtained values. In this section, this correction was used to show the results, and the constraints of measuring graphene are further explored and reported. The test structure used is a lumped element of a 1 kΩ series resistance with 1 NH series inductance, which represents the resistance of a graphene sheet of 1 to 3 squares in length, which accounts for a device with inductive behavior.

### 3.1. Metal Thickness Effect on Calibration Accuracy

As described in [23], when graphene is transferred onto a substrate that contains a large step (such as a measurement contact), if this step is large enough, it may lead to a rupture in the 2D material. Therefore, to obtain a proper contact and avoid damage to graphene, it becomes necessary to fabricate contacts with a small thickness. On the other hand, this causes the contacts to have a large series resistance, which can compromise the reliability of the TRL calibration due to its dependence on transmission line characteristics [16,17].

An example of a graphene contact would be using a 20 nm thick Au contact, given that it allows for a small thickness to prevent graphene from rupturing and allows for a good contact resistance. In this case, to extract the RF characteristics, the TRL calibration method is evaluated. To test the effect of such small thickness on the calibration accuracy, the calibration was performed with 20 nm of Au. Figure 2 shows the results of performing the calibration and extracting the inductance and resistance of the lumped element.

An accurate calibration would yield results close to 1 NH for the imaginary part and 1 kΩ for the real part. However, it is possible to observe that the inductance changes from approximately 0.2 nH to 0.7 nH, and the resistance is around 600 Ω. Therefore, it is possible to conclude that very thin contacts compromise the implementation of the TRL calibration and extraction of the device’s properties.

### 3.2. Substrate Loss Effect on Calibration Accuracy

Since the characteristics of the transmission line can influence the accuracy of the calibration procedure, the substrate conductivity can be another inaccuracy source. To demonstrate the impact of the substrate conductivity, a standard clean room doped silicon wafer with 100 Ω.cm is used. The measurement structures are as previously described, and the contacts are made of 5 µm thick copper to remove the previously described effect of the metallic resistance of the calibration accuracy. Figure 3 shows the comparison between the reference values from the lumped element and the extracted values through the TRL calibration.

It is possible to observe that the substrate loss impacts the extraction of the RF characteristics of the device. The series resistance is in the 800 Ω to 900 Ω range when it should be around 1 kΩ, and the inductance is in the −165 nH to 0.8 nH range when 1 nH is expected. This is due to the increased transmission line losses that are induced by the substrate conductivity.

### 3.3. Effect of Transition from Metallic CPW to 2D Materials

As demonstrated, using thin metallic contacts and lossy substrate cause the calibration to provide very unreliable results. This is due to the extremely high losses that are introduced in the transmission line. To overcome these issues, two approaches must be taken: reduce the contact resistance and reduce the substrate conductivity. To reduce the contact resistance, copper was used, which has one of the highest conductivities, and its thickness was increased to 5 µm to further improve the contact. For the case of substrate conductivity, a commonly available 10 kΩ.cm silicon substrate was used.

The problem of using 5 µm thick contacts is that these will break graphene, and contact will not be made. Therefore, a 20 nm gold contact with a length of 35 µm is included to bridge the connection from the 2D material to the copper contact. Figure 4 depicts two scenarios where a gold contact is used, and a gold contact is not used. For the case where the gold contact is not used, Figure 4A, it is expected that graphene breaks and does not make proper contact with the measurement contact, Figure 4C. For the case where the 20 nm contact is included, Figure 4B, graphene breaks with the height of the copper contact but still makes contact with the gold contact since its height is very small to break graphene. In this case, it may either lay flat, Figure 4D, or roll back, Figure 4E.

Figure 5 displays the results of the calibration when performed with 5 µm thick contacts and 10 kΩ.cm silicon substrate conductivity and the results of the calibration when these structures include the 20 nm gold contact with a length of 35 µm.

It is possible to observe that the accuracy of the measurements has increased when compared with the scenario where the structures have thin contacts, and the scenario where the substrate has a high conductivity. The extracted inductance is now around 0.9 nH to 1.1 nH, and the resistance is between 900 Ω to 1 kΩ, when 1 nH and 1 kΩ were expected, respectively. For example, when comparing with the high conductivity substrate scenario, where the inductance was much lower than 0.6 nH for frequencies lower than 25 GHz and the resistance was between 800 Ω to 900 Ω, this demonstrates that using 5 µm thick copper contacts and 10 kΩ·cm silicon substrate provides a reliable calibration for measuring 2D materials. Furthermore, 20 nm gold contact with the length of 35 µm and with a conductivity of 4.1 × 10^7^ S/m was included. These structures were simulated, and the characteristics of the lumped model were extracted. Figure 5 shows the results of the calibration where the extracted inductance was around 0.9 nH to 1.1 nH and resistance was between 900 Ω to 1 kΩ when 1 nH and 1 kΩ were expected, respectively. It is possible to observe that the impact of including these contacts is negligible and allows us to obtain a working device.

### 3.4. Effect of Backgate on the Extracted Graphene’s Characteristics

Most applications of graphene use its conductivity that depends on its chemical potential. Such property can be modified through a voltage gating and has been explored for tunable applications [24]. Therefore, it is a material that benefits from a characterization for different chemical potentials. This can be achieved through a buried gate that can be included in the described calibration structures. Figure 6 shows the calibration structure used to determine the DUT’s characteristics, with a buried gate to control the graphene’s chemical potential through voltage gating. This structure was simulated and was used to extract the lumped element’s inductance, extracted inductance, and resistance. These values are shown in Figure 7 and have not changed significantly from the previous scenarios, obtaining an extracted inductance of 0.9 nH to 1.1 nH and a resistance from 900 Ω to 1 kΩ, when 1 nH and 1 kΩ were expected, respectively.

### 3.5. TRL Validation for 2D Material Properties Extraction

The previous simulations used a lumped element to mimic the extraction of the properties of graphene. That approach, however, does not account for the signal propagation in the device as well as its interaction with the backgate contact. Therefore, to mimic graphene, the lumped element was replaced by the best possible approach allowed by the simulator—a material with 1 × 10^5^ S/m and a thickness of 20 nm, resulting in 500 Ω/square. The DUT area corresponds to two squares in series, which totals 1 kΩ series resistance. To determine the characteristics of the mimic device, the TRL calibration without the backgate was performed as well as with a backgate made of gold. Figure 8 shows the results of the extraction of the DUT’s characteristics for both cases. It is possible to observe that including a gold backgate impacts the extraction of the device’s characteristics, making it impossible to extract the correct values.

An alternative could be implementing a backgate design that has a high impedance, which would prevent the RF signal propagation from being influenced by the backgate. However, this design would need to be fit for a frequency range from 1 GHz to 65 GHz. Since the purpose of the backgate is transmitting a DC voltage to gate the 2D material, a simpler solution would be to increase its resistance. The only constraint is that the gate oxide must be much more resistive than the series resistance of the backgate itself, so that the applied voltage at the gate is close to the applied voltage at the gate contact. Therefore, the same backgate design can be used if the material’s conductivity is modified. Since the highly conductive backgate contact impacts the parameter extraction, by reducing the backgate conductivity, it is possible to circumvent this problem. The extraction structures, with the same backgate design were simulated with a material with a conductivity of 2 × 10^4^ S/m. Figure 9 shows the result of the simulation with the low conductivity backgate and the result without the backgate. It is possible to conclude that to obtain an accurate extraction method, a low conductivity backgate is necessary. Alternatively, to accommodate the structure to different materials, there is still room to change the backgate dimensions, such as using a thinner backgate contact or reducing the contact width.

## 4. Measurement of Graphene’s Characteristics

The proposed measurement and calibration structures were fabricated. In this case, the 2D material was graphene, and the device fabricated is shown in Figure 10. It is possible to observe the phenomenon reported in Figure 4, where the graphene breaks due to the large contact height, and the thin gold contacts bridge the contact between graphene and the copper contacts. Furthermore, after breaking, graphene also rolls back in some cases. In this structure, the height of the copper contacts is 5 µm, and the length of the gold contacts is 35 µm. The gated area of graphene has a length of 40 µm and width of 50 µm, where a meander was used to increase the gate resistance. This meander is 3 µm wide and has 24 turns, with a total length of 20,827 µm and a contacting 2191 µm interconnect length, resulting in a total length of 23,018 µm.

The fabricated structures were measured with the Keysight E5071C VNA (Keysight technologies, Santa Clara, CA, USA), the GSG |Z|-Probes (FormFactor, Livermore, CA, USA), and the CS-5 calibration substrate (GGB Industries, Naples, FL, USA). Then, the previously proposed calibration procedure was undertaken, allowing us to extract graphene’s characteristics. These measurements were performed for 0 V, 5 V, 10 V, and 15 V of backgate bias. The extracted resistance and quantum capacitance of graphene are shown in Figure 11. This data was fitted and plotted in red lines to allow for an easier interpretation.

The extracted resistance changes from around 450 Ω at 1 GHz down to around 250 Ω at 8.5 GHz for the bias voltage of 0 V. It then increases with the increase of the backgate voltage up to a maximum resistance of around 750 Ω at 1 GHz to around 400 Ω at 8.5 GHz, obtained at 15 V. The quantum capacitance changes from close to 0.55 pF at 1 GHz down to around 0.2 pF at 8.5 GHz for the bias voltage of 0 V. This capacitance reduces with the increase in voltage, achieving a capacitance of around 0.23 pF at 1 GHz down to around 0.1 pF at 8.5 GHz for the bias voltage of 0 V. Both resistive and capacitive behavior are as reported in the literature [25,26,27]. The fabrication process used yields a typical mobility from 1000 to 2000 cm^2^V^−1^s^−1^, and the Dirac point is in the range of 6 to 8 V when it is gated with 15 nm of Al2O3, meaning that in this case, it should be from 50 to 67 V.

## 5. Conclusions

Graphene is a promising material for several fields such as electronic and RF applications. The TRL calibration provides one of the highest calibration accuracies. In this work, the application of the TRL calibration for the extraction of the RF characteristics of graphene was explored. The implementation of the calibration was described, as well as the parameters that negatively affect its accuracy such as thin contacts and lossy substrate. To obtain an accurate calibration, thick contacts and low loss substrate should be used to reduce the losses in the transmission line. In this case, considering the characteristics of the transmission line, the series resistance is around 0.8 Ω, the substrate conductivity 10 kΩ·cm, and the backgate contact around 730 kΩ, which can be used as a reference point for the calibration. Since thick contacts will damage graphene, a short length thin contact was added, and its impact on the extraction of the RF characteristics of graphene was demonstrated to be negligible. Furthermore, since graphene’s conductivity changes according to its chemical potential, a backgate was added to characterize it for different potentials. Finally, the proposed structures were fabricated and measured, and the properties of graphene were extracted, from 1 GHz to 8.5 GHz. Its resistance at 1 GHz changed from around 450 Ω to 750 Ω at 0 V and 15 V, respectively; at 8.5 GHz, it changed from around 250 Ω to 400 Ω at 0 V and 15 V, respectively. Its quantum capacitance at 1 GHz changed from around 0.55 pF to 0.23 pF at 0 V and 15 V, respectively; at 8.5 GHz, it changed from around 0.2 pF to 0.1 pF at 0 V and 15 V, respectively. The development of such calibration procedure to characterize graphene will allow us to further improve the knowledge of the graphene’s RF characteristics, paving the way to obtain tunable electronics either through biasing or biological functionalization. Finally, the proposed calibration and extraction method are not limited to graphene and can be used with other 2D materials.

## Figures and Tables

**Figure 1 micromachines-14-00215-f001:**
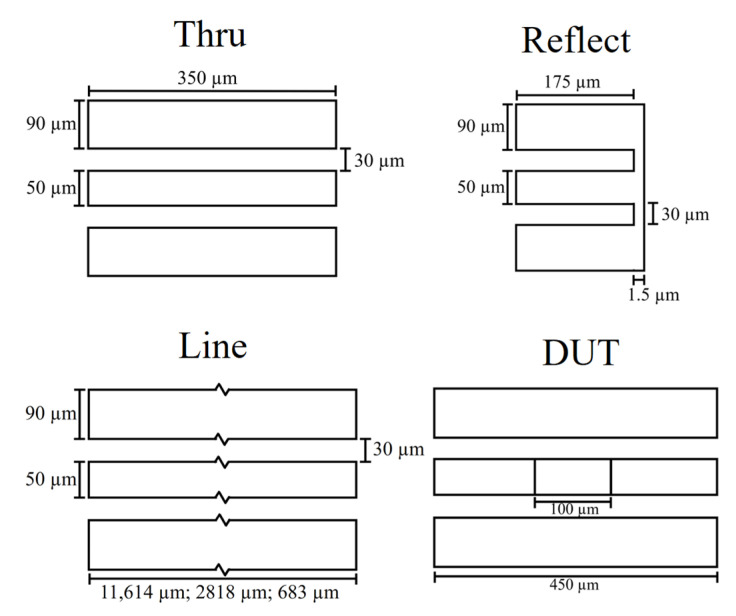
Calibration structures and dimensions used in the implementation of the TRL calibration algorithm.

**Figure 2 micromachines-14-00215-f002:**
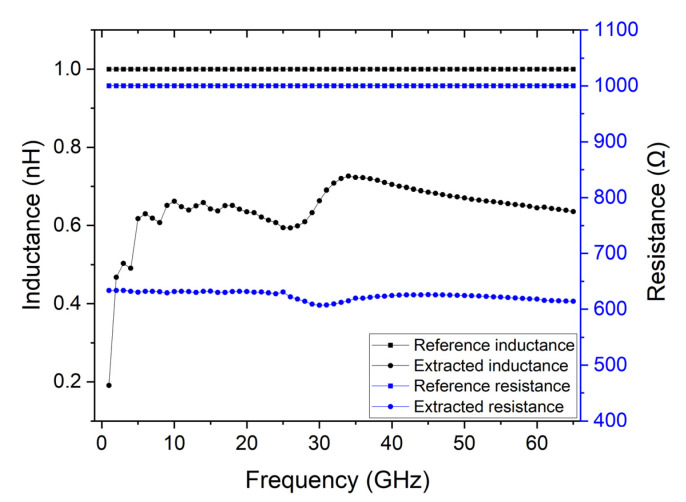
Comparison of extracted inductance and resistance parameters of the lumped model when 20 nm thick gold contacts are used, and the expected reference values.

**Figure 3 micromachines-14-00215-f003:**
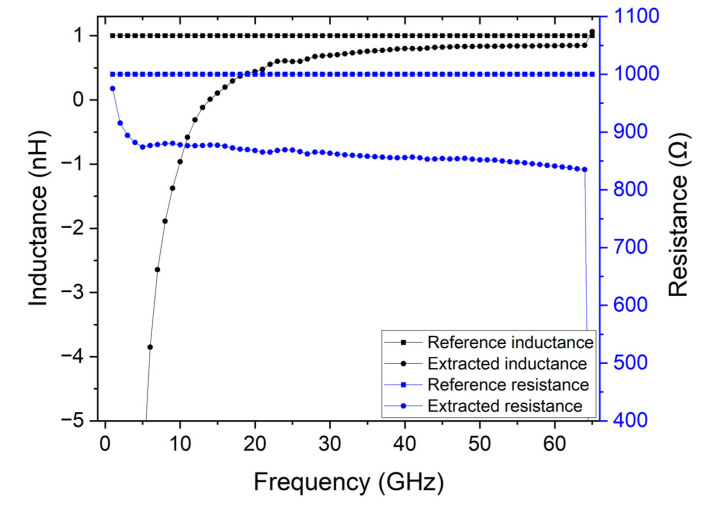
Comparison of extracted inductance and resistance parameters of the lumped model when a substrate with 100 Ω.cm conductivity is used and the expected reference values.

**Figure 4 micromachines-14-00215-f004:**
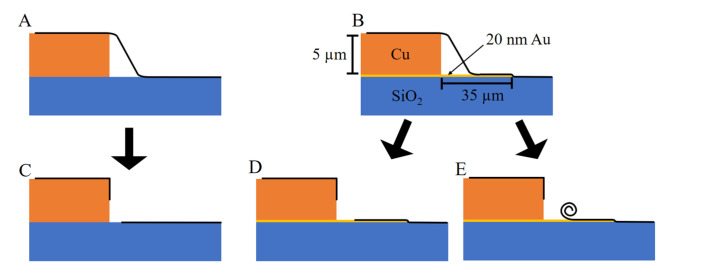
Metallic contact between copper and the 2D material: (**A**) the case where the material is contacted directly to copper; (**B**) the case where the material is contacted with a 20 nm thick gold contact; (**C**) the consequence of undertaking scenario A and demonstrating the 2D material breaking and not making contact with copper; in (**D**) the consequence of scenario B, where the contact between the copper and 2D material is mediated through a 20 nm thick gold contact and the 2D material lays flat; (**E**) the consequence of scenario B, where the contact between the copper and 2D material is mediated through a 20 nm thick gold contact, and the material breaks and rolls back.

**Figure 5 micromachines-14-00215-f005:**
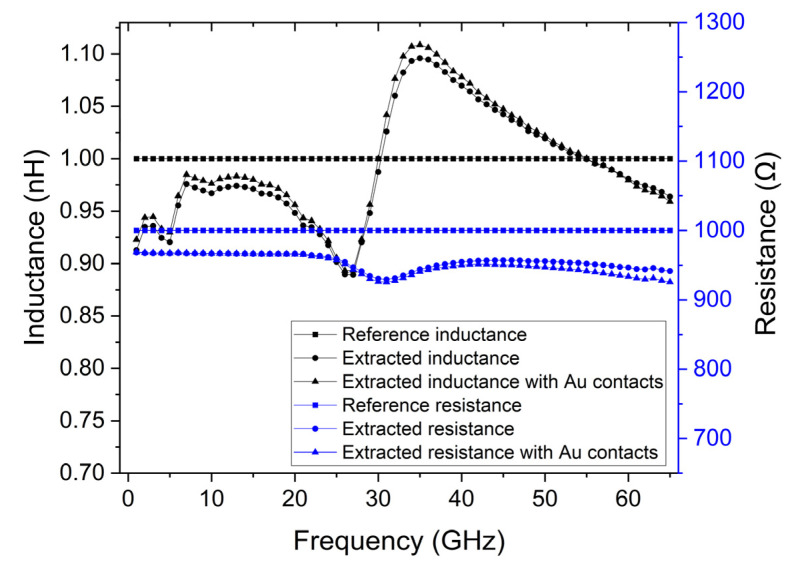
Comparison of extracted inductance and resistance parameters of the lumped model when the contacts are made of 5 µm thick copper and the substrate has a conductivity of 10 kΩ·cm, and when 20 nm thick gold contacts are included, with the expected reference values.

**Figure 6 micromachines-14-00215-f006:**
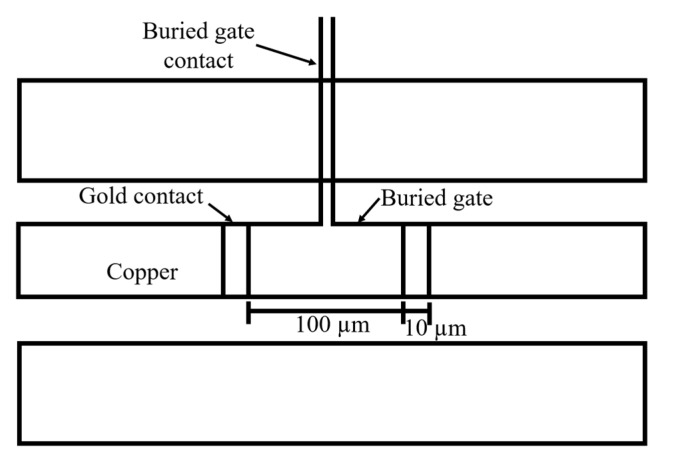
Diagram of the DUT measurement structure with the buried gate, with a contact width of 3 µm, length of 437 µm, and a square contact pad with a side of 150 µm.

**Figure 7 micromachines-14-00215-f007:**
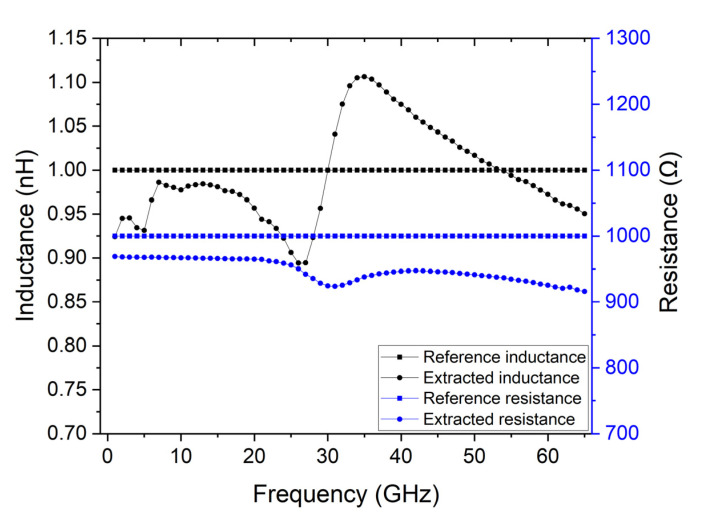
Comparison of the lumped element’s inductance and resistance and the extracted parameters of the lumped model when the contacts are made of 5 µm thick copper, the substrate has a conductivity of 10 kΩ.cm, 20 nm thick gold contacts are included, and a 10 nm thick back gate is introduced.

**Figure 8 micromachines-14-00215-f008:**
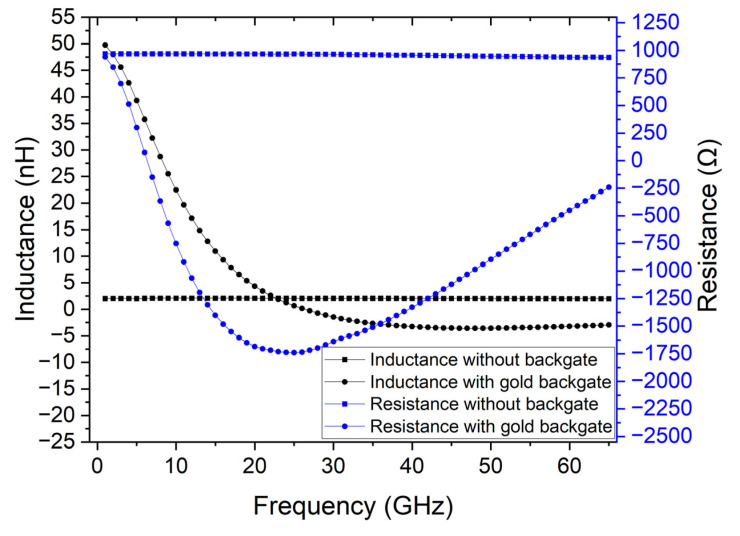
Comparison of the simulated and extracted DUT’s characteristics without the backgate and with a 10 nm thick gold backgate.

**Figure 9 micromachines-14-00215-f009:**
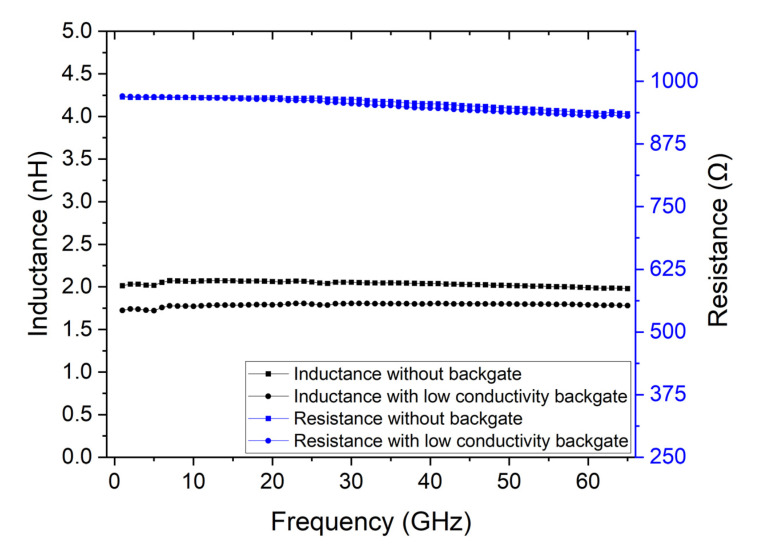
Comparison of the simulated and extracted DUT’s characteristics without the backgate and with a 10 nm thick backgate made of custom material with a conductivity of 2 × 10^4^ S/m.

**Figure 10 micromachines-14-00215-f010:**
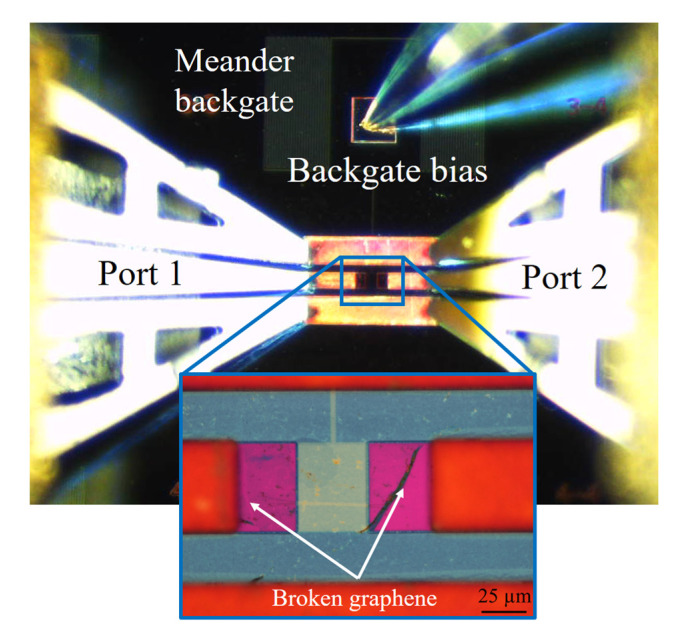
Fabricated device for graphene characterization.

**Figure 11 micromachines-14-00215-f011:**
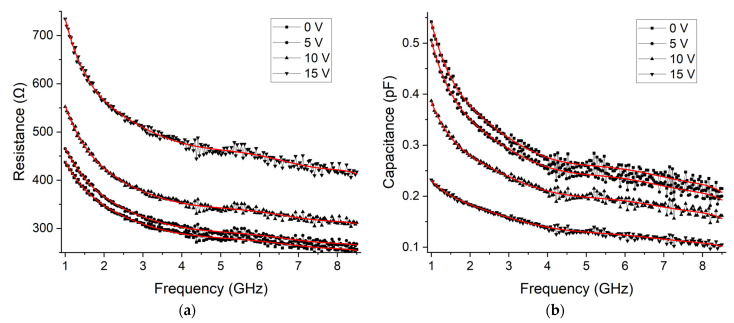
Extracted resistance (**a**) and quantum capacitance (**b**) of graphene for backgate bias voltages of 0 V, 5 V, 10 V, and 15 V. Fitted lines are shown in red for an easier interpretation of the results.

## Data Availability

The data presented in this study are available on request from the corresponding author.

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
