# Peer review of "Extraction of Graphene’s RF Impedance through Thru-Reflect-Line Calibration"

_micromachines, 2023, doi:10.3390/mi14010215_

Round 1

Reviewer 1 Report

It is indisputable that graphene can have an important role in the RF applications field and the proposal of accurate calibration methods to extract as precise as possible its performance characteristics is relevant and necessary. The work by Colmiais et al is interesting. The authors develop in detail the simulation methods, the geometry of the device components and the calculation procedures. In general, they explain how the dimensions of the different components should be in order to optimize the calibration protocol and they also analyze the results. Finally, the fabrication of a real device and the experimental results provided to test the proposed method and to compare with the simulations is a positive point.

For these described reasons, the manuscript is in principle suitable to be accepted in the journal Micromachines. However, it should be amended in several aspects before the publication:

- The message “Error! Reference source not found” is written 18 times along the text, in the different sections. This is not serious when a final version of a submitted manuscript is expected. Sometimes, due to this message, the text that is next to it is unintelligible. Please, correct these errors and add the right information.

-The captions in figure 4 are difficult to understand in the current form. Please, amend these captions.

-Please, in figure 5: could the authors explain the appearance of peaks around 25 and 35 GHz in the graphs of the extracted inductances? I ask the same question with the appearance of the peaks in the graph of the extracted inductance in figure 7. I ask the same question with the observed tendency of the inductance and the resistance between 20 to 30 GHz in figure 8.

-In the simulations, the length of the gold contacts used was 10 microns (lines 256-257). However, in the experimental section, for the real device, the used length of the gold contacts was of 35 microns (line 330). Could the authors explain where this difference comes from?

- In the simulations, the authors study the inductance and the resistance in the range of 1 to 65 GHz. However, in section “Measurement of Graphene’s Characteristics”, with the real measured device, the authors provide results within a range of 1 to 8.5 GHz. Why is this? Then, how could the accuracy of the proposed TRL calibration method be checked in the range from 8.5 to 65 GHz?

- In the introduction section was already stated, and the last sentence of the conclusion section claims that the proposed calibration and extraction method could be used with other 2D materials. 2D materials share common traits undoubtedly. However, they can present very different characteristics and behaviour as well, depending on the specific 2D material. Please, the authors should elaborate more in detail this hypothesis.

Finally, other points that are not so scientific-related but it would be nice if taken into account:

- Please, put the definition of a term and its acronym between parenthesis the first time the term is mentioned in the text such as: device under test (DUT). This, along the text, several times, with the terms: VNA, SOLT, LRM, TRL, coplanar waveguides. Actually, in the title, I would put the full definition of the term TRL, such as: “Extraction of graphene’s RF impedance through Thru-Reflect-Line calibration”

-In line 128, when the authors write “through equations 8 and 9”, I believe they refer to the equations 5 and 6.

Reviewer 2 Report

This paper presents a method based on TRL calibration to extract the RF impedance of graphene using TRL calibration. The paper matches well the objetive with the theory, fabrication and validation throughout the measurements.

I have few remarks that authors must address:

(1) it is said that graphene can rolling back when it breaks. This can be shown and illustrated in figure 4.

(2) in figure 4, C is the consequence of A, as D is consequence of B. This in not clear and can be improved by adding arrows and tags when applicable.

(3) it is stated in line 331 the use of a meander to increase the gate resistance; in this sequence, at least their dimensions could be presented.

(4) fix the exponents on “conductivity of 4.1 x 107 S/m” in line 257, where the “7” is the exponent; this happens several times allong the manuscript.

(5) increase the space between the captions and the images.

(6) increase slightly the letter size on figure 10.

Round 2

Reviewer 1 Report

The authors have kindly addressed point by point all my questions. they have replied convincingly and they have amended my suggestions. Therefore, I believe the manuscript is ready to be accepted in the journal Micromachines.